# The self prefers itself? Self-referential versus parental standards in face attractiveness

Unni Sulutvedt and Bruno Laeng

Department of Psychology, University of Oslo, Norway

## ABSTRACT

Preference for phenotypic similarity in assortative mating may be influenced by either a preference for self-similarity or parent-similarity. The aim of the current study was to assess whether people's preference in face attractiveness is influenced by resemblance to the opposite sex parent's face (parental phenotype) or their own face (self-based phenotype or "self-imprinting"). We used computerized image manipulations of facial photographs of participants, their mothers and fathers. The original photographs were combined with 78% of the participants' opposite sex prototype face (i.e., male and female prototypes made from equal contributions of a hundred faces), creating morphs where the contribution from the familiar faces went unnoticed. Female and male participants ranked these images together with the opposite-sex prototype different familiar morphs. Each participant did the same for the morphs generated with other same-sex participants' faces and of their parents. We found that the female participants preferred the self-based morphs to the prototype faces. Male participants showed a general tendency towards self-referential standard. Parental face morphs were ranked low in attractiveness, which may be accounted for by the age difference of the faces blended into the self-based versus parental face morphs, since we used present-time photographs of both the participants and their parents.

## INTRODUCTION

According to current evolutionary thinking, a recognition process based on phenotype matching to specific individuals during early childhood provide the basis for whom will be considered a potential sexual mate later in life. *Bateson* (*1978*, p. 265) postulated an "optimal outbreeding" mechanism according to which "sexual imprinting sets the standard (or standards) of what immediate kin look like and the animals subsequently prefer to mate with an individual who looks slightly different". He also surmised that sexual imprinting exists in humans and that self-stimulation with own phenotype (e.g., self's odor for some animals) may get sexually imprinted. Indeed, several researchers have used the term sexual or familial "imprinting" during childhood to describe an early learning process, often not in a strict Lorenzian sense (*Lorenz, 1965*) but as an "imprinting-like" type of mechanisms (e.g., *Little et al., 2003*) where early experiences may have a primacy

Corresponding author
Unni Sulutvedt,
unni.sulutvedt@psykologi.uio.no

over later ones so that exposure to individuals that happen to be in closest proximity during childhood (typically, close kin) are most likely to provide "templates" for later mate choice (see also *Lieberman, Tooby & Cosmides, 2007*). Such a kin recognition process is also at the foundation for seeking a balance between incest avoidance (cf. *Shepher, 1971*) and an opposite tendency to seek family resemblances.

In addition, *Symons (1995)* observed that, in current human environments, mirrors are ubiquitous and they may affect attractiveness perception (particularly of same-sex individuals) in evolutionarily unprecedented ways, and may even be responsible for some positive assortative mating *Symons* (*1995*; p. 108). However, he also stressed the possibility that already during the course of human evolutionary history our ancestors could see their reflections in still water. Hence, the historically recent spread of use of mirrors could have boosted the effects of an already selected "self-imprinting" mechanism and humans did not need to wait for the appearance of mirrors to get knowledge of self-appearance and be influenced by it. In fact, still water is highly reflective (consider the myth, at least 2,000 years old—of Narcissus falling in love with his image in a pond) and reflective surfaces like still water have been present since the beginning of evolutionary time. Thus, the potential for self-imprinting clearly predates the mass production of mirrors, as well as that of portraits, homemade photos (e.g., "selfies") and movies.

Remarkably, people tend to marry individuals who are similar to themselves. This phenomenon has been widely documented within Western societies (e.g., *Alvarez & Jaffe, 2004*; *Bereczkei et al., 2002*; *Zajonc et al., 1987*), where individuals tend to pair and wed willingly and not that commonly in pre-arranged marriages. Some of these studies showed that when participants were asked to pair pictures of unknown individuals of both sexes (*Zajonc et al., 1987*), photos of actual partners were paired above chance. Moreover, *Alvarez & Jaffe (2004)* found no difference between the matching of the ones most likely to be siblings and the ones most likely to be married, which indicates the existence of a high degree of similarity between partners. Finally, self-perception appears to modulate mate preference (*Buston & Emlen, 2003*), which is consistent with the hypothesis that human's criterion for beauty is rooted on an image of self. These findings support the existence of a detection mechanism of similarity to the self that influences attraction to others' faces.

Regarding mate choices, "imprinting" to the father's face has been shown to be relevant for the daughter's facial preference (e.g., *Wilson & Barrett, 1987*; *Bereczkei et al., 2002*; *Little et al., 2003*; *Watkins et al., 2011*; *Wiszewska, Pawlowski & Boothroyd, 2007*). For example, females found facial stimuli that resembled their father more attractive if they had a good relationship with their father during childhood (e.g., *Wiszewska, Pawlowski & Boothroyd, 2007*). However, this imprinting-like process does not exclude that also the self's face may have an impact on the formation of the kin template. Also, the child's attachment to the mother figure and early exposure to the mother's face is essential in the social development of humans (*Bowlby, 1969*; *Belsky, Steinberg & Draper, 1991*) and this could also influence later affiliative choices. One possibility, which will be specifically explored in the present study, is that experiences of faces in close proximity during childhood could influence, at a later stage in life, adults towards a "parental/kin standard" of attractiveness, featuring the

facial characteristics of self's parents. In addition, self-referential standards could become established over a lifetime exposure with the self's own face and in fact reinforce kin-based preferences.

It is likely that several early "face templates" are shaped by exposure to faces throughout childhood which leave a lasting impression on an individual's face processing mechanisms (*Perrett et al., 2002*) and in turn influence mate choices in adulthood. Studies on facial attractiveness are abundant and they have revealed that both the symmetry and averageness of a face are crucial for attractiveness. However, if these were the sole perceptual factors affecting people's sense of facial beauty, then most people will be most attracted to the average face (e.g., a prototype; *Halberstadt & Rhodes, 2000*).

A crucial assumption for the present study is that any deviation from the preference for the prototype would support the idea that there exist important individual differences in the sense of face aesthetics. Also, if facial preference is reduced to just a "mere exposure" phenomenon (*Hill, 1978*; *Moreland & Zajonc, 1982*), where highly familiar faces are regarded as more likeable and attractive, then faces resembling any of the family faces (e.g., mother, father, as well as self) should be found to be equally attractive and possibly preferred to the opposite-sex prototypes. To our knowledge, attempts to separate a parental criterion from a self-referential have never been tested by requesting the same group of participants to compare simultaneously faces similar to themselves and their parents.

## Present study

At the basis of the present research is evidence that lovers—when given the opportunity—would prefer their partners to resemble themselves (*Laeng, Vermeer & Sulutvedt, 2013*). Specifically, a 22% self-based morph with the partner's face was preferred over the morph of the partner's face with its same-sex's prototype. Thus, when given the opportunity, partners will make aesthetic choices towards increased resemblance to self, either for the whole face or some distinctive elements (e.g., the eyes' color; Laeng, Mathisen & Johnsen, 2007). Nevertheless, it remains unclear whether these findings reflect a self-referential mechanism or a parental mechanism or a combination of the two. Thus, the present study attempted to tease apart the contribution of familial imprinting to phenotype matching to self. To our knowledge, this is the first time that the latter effects have been studied with graphic manipulation methods based on the face images of the participants' parents.

Since the learning process of kin detection and imprinting may influence both sexual and non-sexual preferences (e.g., trust or friendship), the effects of family resemblance may be better captured by asking participants to rank images according to their "attractiveness", without explicitly mentioning sexual attractiveness. Thus, we used computerized image manipulations, morphs, of facial photographs of self, and self's biological mother and father in order to assess this dilemma. That is, stimuli were made by a weighted average of every pixel in the two images, so that all aspects of the faces (shape and pigment) were moved towards an intermediate representation which was to all effect a novel face stimulus. We used the opposite-sex prototype face (of the participant's age cohort) as the base image into which the same small amount of information (22%) of self, father, and mother were

morphed. All images were presented simultaneously as high-quality color prints and participants were asked to rank them for their attractiveness. In addition, the same images from another subject's protocol were presented for comparisons between *Family* and *Other*. The opposite-sex prototype face (i.e., the morphed average of 100 people of the participants' age-cohort) was also presented, so as to provide a baseline for comparing increase or decreases in the sense of beauty due to the other image manipulations.

We used morphs with a blending of 22% because they have previously been shown, by use of "objective criteria of awareness" based on forced-choice tasks and signal detection analysis (Merikle, Smilek & Eastwood, 2001), to be perceived below consciousness, as opposed to morphs with larger contributions, e.g., 33% (*Laeng, Vermeer & Sulutvedt, 2013*). Moreover, *Laeng, Vermeer & Sulutvedt (2013)* found that attractiveness judgments on self-morphs combined with attractive strangers peaked at 22%, relative to 11% or 33% morphs. This may indicate that 22% approximates the optimal amount of likeness in order to prevent avoidance mechanisms of primary incest. As seen in a pilot study, this 22% peak effect was not observed if the stranger's face was previously rated as unattractive, in which case other evolutionary "constraints" may dominate—e.g., the variations in asymmetry of individual faces (cf. *Rhodes, 2006*; *Miller & Todd, 1998*). Therefore, when morphing, we used the opposite-sex prototype faces, in order to achieve a face of optimal averageness and symmetry.

We need to stress that we are not suggesting that average faces are optimally attractive but only that average facial configurations are more attractive than most faces (*Rhodes, 2006*). The logic behind our study is simply based on the well-documented fact that the symmetrical averaged face obtained when morphing multiple faces typically score very high in attractiveness and when adding to a prototype a sizable proportion of a randomly chosen face, this does not cause an increase of attractiveness but typically reduces it a bit. Based on the above evidence, we hypothesize that adding to a prototype a 22% of oneself or kin should result in no gain in attractiveness unless self's appearance or kin's are actually "liked".

The predictions are quite straightforward: If the preference for the phenotype is strongly based on a self-referential process the self-morph should be preferred to all other morphs. In contrast, the self-morph should be ranked as less attractive than the morphs based on kin's faces if parental imprinting plays an important role. Since our participants were heterosexual, we also predicted that the maternal morph should be preferred to the paternal by male participants while the reverse should happen for the female participants. Finally, if facial preference is caused by a "mere exposure" phenomenon, all the morphs based on the familiar faces (e.g., mother, father, as well as self) should be found to be equally attractive.

However, we need to point out from the outset a caveat of the present experiment: we were able to use only 'present-time' images of the participant's parents, which may not be ideal for studying the influence of parental effects, since parents do not look exactly the same today as they did when a parental standard could have been established. Nevertheless, the present study is an initial and direct test of the parental-template hypothesis. Since a
few previous studies have also shown that individuals' preferences in attractiveness may be influenced by attachment experiences in adults' upbringing (e.g., *Wiszewska, Pawlowski & Boothroyd, 2007*; *Nojo, Tamura & Ihara, 2012*), we also used the short form of the EMBU scale (*Arrindell et al., 1999*) to assess positive versus negative attachment to the mother and the father figure.

## MATERIALS AND METHODS

### Participants

The study included a total of 36 heterosexual participants (18 females) between the age of 18 and 33 with normal or corrected to normal vision (mean age = 25.50; $SD = 4.24$). The participants were recruited in the Oslo area, either at seminars at the Department of Psychology or by word of mouth.

Psychological research in Norway is subject to ethical review by the regional medical research board only if the research involves patients, children or animals and involves drugs, genetic samples or invasive techniques. Since none of these conditions applied to the present study, the academic institution demanded only that the project comply with Declaration of Helsinki guidelines and that informed consent be obtained from the participants. We obtained written consent from all participants, and participants were free to withdraw from the project.

### Stimuli and apparatus

Standardized frontal view photographs were taken of the faces of all the 36 participants and their biological mothers and fathers. A total of 108 photographs, using a Panasonic Lumix DMC digital camera, were collected. When parents lived far away, high quality digital photographs were sent by e-mail. The photographs were taken from a distance of 1 m, and participants and their parents were asked to display a neutral, emotionless facial expression. Facial hair was removed and deep creases, folds, and lines of the skin were retouched using Adobe Photoshop© to reduce age effects. We chose to use pictures of parents from present time, after failing to collect pictures of parents from the time their children were young. The old pictures were of too poor quality and not sufficiently standardized (e.g., some showed smiling faces, eye-glasses, different poses, etc.), so that morphing these old pictures with the high quality prototypes would not yield satisfactory images. Hence, we opted to using current photos taken with the same digital camera and in standard pose, and with neutral expressions.

Using image-morphing software (Morpheus Photo Morpher©), 22% of all the original photographs were blended in with the opposite sex prototype face of the participant's age cohort (made from 100 faces) to create self-referential, paternal, and maternal morphs (see Fig. 1). On all images, 71 points were placed on different features of the faces before morphing. As in *Lie, Rhodes & Simmons (2008)* these color morphs (239 × 316 cm) were digitally fitted with an oval black mask (not shown in the figure) to reduce any effects of clothing and hairstyle, leaving only the face and minimal hair visible.

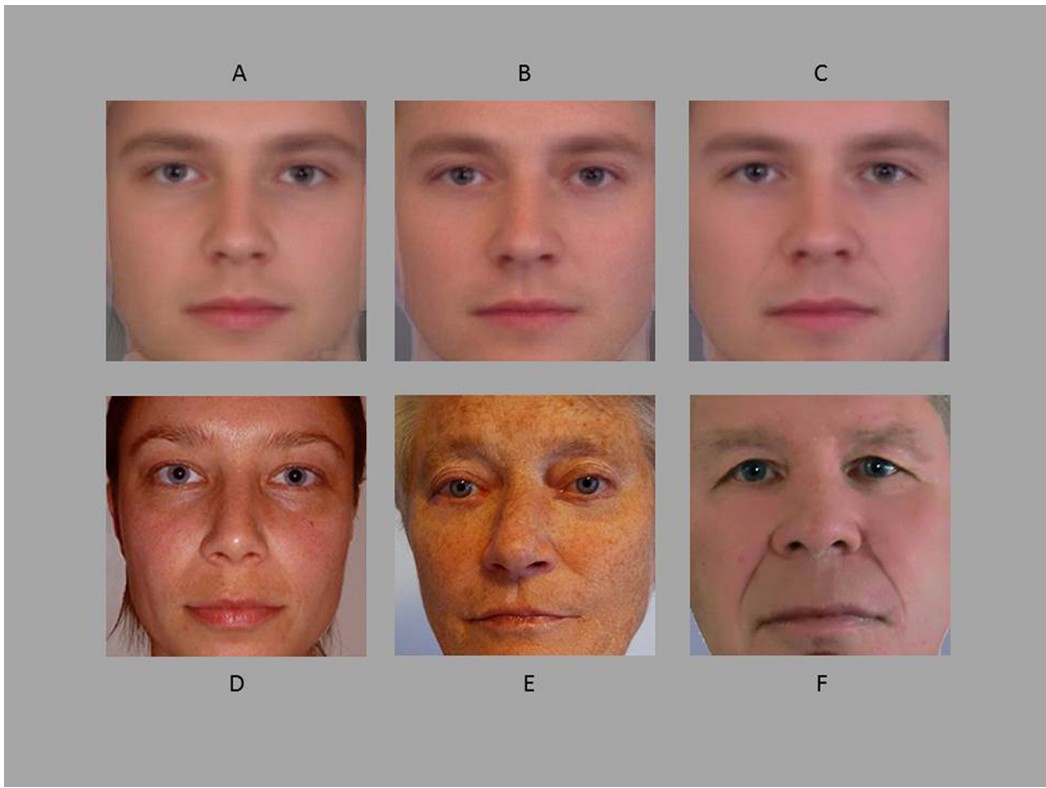

**Figure 1 Examples of morphed image-set and original images.** Top row: Examples of the morphed image-set based on the author: (A) Self, (B) Mother, and (C) Father. Bottom row: Original images of the author (D) Self, (E) Mother, and (F) Father.

The s-EMBU retrospective attachment test (*Arrindell et al., 1999*; *Arrindell et al., 2001*) was used to control for perceived parental attachment in childhood. The s-EMBU is a 23-item inventory with 4-point Likert type scales, to be answered separately for mother and father. This test consists of three scales: Emotional Warmth, Rejection, and (Over) Protection each of which comprises eight items. Following *Gyuris, Járai & Bereczkei (2010)* only Emotional Warmth and Rejection was used to assess perceived positive and negative emotions towards parents during upbringing.

## Procedure

Each participant was matched by age to another participant of the same sex so as to obtain a set of morphs for the "Other" condition. Four morphs of were presented side by side for each separate sessions for "Family" or "Other" conditions. The participants were asked to rank them from the most attractive (i.e., 1st place) to the least (i.e., 3rd place). The order of the images was random within each block. All the stimulus images were presented centered on the computer screen with a resolution of 1024 × 768 pixels. The software "Experiment Center" by SensoMotoric Instruments® (SMI, Teltow, Germany) was used to present the stimuli.

**Table 1  Mean ranks for Family morphs.**

| Female participants | Mean rank |
|---|---|
| Self morph | 1.53 |
| Male prototype | 2.36 |
| Self's mother morph | 2.97 |
| Self's father morph | 3.14 |

| Male participants | Mean rank |
|---|---|
| Female prototype | 1.89 |
| Self morph | 2.33 |
| Self's mother morph | 2.72 |
| Self's father morph | 3.06 |

Participants were tested individually and were seated in front of a computer screen at a distance of 60 cm. A blank screen was presented for 500 ms followed by a fixation cross presented for 1000 ms and then the image-set was viewed as long as it took to make a decision, and the participants pressed space on the keyboard to move on to the next image-set. The experimenter manually recorded the attractiveness responses (hence we did not analyze response times), and the experiment lasted approximately 30 min, followed by the short questionnaire. After the experiment all the participants were debriefed and asked if they had recognized any of the faces. Only two participants thought they might have recognized their opposite-sex parent, and said they had consequently ranked these morphs in 3rd place. However, removing these participants from the analyses did not affect the overall results and, therefore, we present below results from all participants' responses.

## Results

All data were analyzed using StatView® statistical package. $t$-tests for independent samples were conducted on the s-EMBU to compare differences between sexes in perceived parental behavior. There were no significant differences in the ratings between females and males in any of the elements in the s-EMBU; mother rejection ($p = .68$), father rejection ($p = .32$), mother emotional warmth ($p = .57$), and father emotional warmth ($p = .40$). All participants were high on parental emotional warmth and low on parental rejection, thus, the s-EMBU was not used in further analyses.

Descriptive statistics were calculated for each participant, obtaining mean attractiveness ranks for each combination of the variables (Tables 1 and 2). Rankings from female and male participants were analyzed separately, since the parental hypothesis predicts opposing differences in attractiveness between parental sexes. The attractiveness rankings were analyzed using the Friedman's Rank Test and pair-wise comparisons were carried out with the Paired Sign Test.

The analysis of the female participants in the Family group showed significant difference in ranking of morphs: $\chi 2 = 17.23$, $df = 3$, $p < .0005$ (see Table 1, top). The self−morph was ranked first and the male prototype was ranked second place, these differed significantly from each other according to a Paired Sign Test ($p < .03$), confirming the

**Table 2 Mean ranks for Other morphs (i.e., non-familiar faces).** The 'Other' morphs were the same images labeled in Table 1 as 'Self' morphs.

| Female observers | Mean rank |
| --- | --- |
| Other morph | 1.93 |
| Male prototype | 2.15 |
| Other's mother morph | 2.84 |
| Other's father morph | 3.08 |

| Male observers | Mean rank |
| --- | --- |
| Female prototype | 1.50 |
| Other morph | 2.50 |
| Other's mother morph | 2.58 |
| Other's father morph | 3.42 |

presence of a self-referential preference over and above a preference for average and symmetric faces of the opposite sex. The self-morph also differed significantly from both the mother ($p < .002$) and father ($p < .006$) morphs. The prototype was preferred significantly to the mother morph ($p = .049$). Interestingly, comparing the male prototype and the father's morph showed no significant difference between their ranks ($p = .21$), which shows that the parental face did not invariably resulted in an unattractive face morph.

When the same female participants judged morphs in the Other set, they also showed significant differences in their ranking of attractiveness; $\chi2 = 9.22$, $df = 3$, $p < .03$ (see Table 2, top). However, the difference between the Other morph (i.e., the morph replacing the self-morph of the above analyses, Table 2) and the male prototype failed to be significant ($p = .212$), which confirms that only when the morph contains a bit of self-resemblance such a morph is preferred over the prototype. The prototype was significantly preferred to both the Other's mother morph ($p = .03$) and the Other's father morph ($p = .001$). Finally, Other was preferred to both Other's mother morph ($p = .005$) and the Other's father morph ($p = .052$). There was no difference in the ranks of the Other's mother morph versus the Other's father morph ($p = .091$).

Analysis of male participants in the Family group revealed the presence of a a significant difference among ranks, $\chi2 = 8,20$, $df = 3$, $p = .04$ (see Table 1, bottom). This was entirely accounted, according to the Paired Sign Tests, by a significant difference for the morph ranked first versus the one ranked last, i.e. the female prototype over the father morph ($p = .041$). When the same male participants judged morphs in the Other set, they showed significant differences in their ranking of attractiveness, $\chi2 = 19.95$, $df = 3$, $p < .0001$ (see Table 2, bottom). They clearly preferred the female prototype over the Other morph ($p = .0013$) as well as the Other's father ($p = .0001$) or mother morphs ($p = .01$). Moreover, the Other morph was significantly preferred to the Other's father morph ($p = .0001$) as well as to the Other's mother morph ($p = .01$).

## DISCUSSION

The present results are consistent with the hypothesis that people possess a self-referential standard of face attractiveness, since self-based morphs were preferred to the prototype morphs, especially by the female participants. They are also in line with several previous findings of "narcissistic" preferences for faces (e.g., *Alvarez & Jaffe, 2004*; *Bereczkei et al., 2002*; *Zajonc et al., 1987*; *Laeng, Vermeer & Sulutvedt, 2013*). Interestingly, a previous study with morphs (*Laeng, Vermeer & Sulutvedt, 2013*) also revealed stronger preferences for self-morphs in females than in males. It is possible that these effects may be caused by a greater tolerance for same-sex blends (i.e., feminization) with female participants than for the males (i.e., masculinization). Another possibility is that females are more exposed on average than males to the self's phenotype by use of mirrors. In the present study, the male participants' choices were also consistent with a self-referential standard, since for male participants the female prototype did not differ significantly in attractiveness from the self-morph.

However, we failed to show evidence in favor of the parental template hypothesis or family imprinting hypothesis, because the self-morph was preferred over the maternal and paternal morphs and the opposite-sex prototype was preferred over both parental morphs. In addition, the Other morphs were ranked above their parents' morphs. When taken together these results suggest that there was an effect of the age of the face morphed into the prototype, where the older ages of parents to both self's and other's reduced the attractiveness of all parental morphs. Clearly, a caveat of the present experiment is that it used 'present-time' images of parents which may have counter-acted the influence of possible parental effects, since parents do not look exactly the same today as they did when a possible parental standard could have been "imprinted". Although we digitally retouched skin creases, folds, and lines to reduce the cues of age, the wide age difference may still create confounds in the images. If a parents' template is established during the first years of life, individuals might actually be mostly attracted to faces resembling what the parent looked like at the time self was a child.

However, it is interesting to note that while the females significantly preferred the male prototype to both Other's mother and father morphs, this difference disappeared when the male prototype was compared to the self's father morph. This suggests that family resemblance to the opposite-sex parent may at least mitigate the negative "age effect" on females' preference. A few previous studies have found the presence of facial similarities between the faces of the spouse and the parent of the opposite sex (*Little et al., 2003*; *Marcinkowska & Rantala, 2012*; *Wilson & Barrett, 1987*; *Wiszewska, Pawlowski & Boothroyd, 2007*). One study on facial similarity between spouses and their parent (*Nojo, Tamura & Ihara, 2012*) used images of parents from when the participants were children, and found no clear implications of the similarity between the couples and their mother/father-in-law. Thus, the present results only allow us to draw tentative conclusions about parental versus self-referential tendencies in face attractiveness, since it is difficult to disentangle the effects due to the age of the face morphed into the prototype from that of its family status and the present results are largely consistent with younger faces being preferred to older ones.

In addition, we found no significant differences in ratings of perceived Emotional Warmth or Rejection from fathers or mothers during upbringing. All the participants had quite similar mean scores, as both parents were rated rather high on Emotional Warmth and low on Rejection. Despite the participants' seemingly good relationship to their parents, the self-resembling faces were chosen over the parents faces, suggesting that positive feelings towards parents do not indicate a greater parental tendency in face attraction. *Wiszewska, Pawlowski & Boothroyd (2007)* compared facial proportions of fathers' faces to the proportions of the facial stimuli chosen by females as the most attractive. They found that there was no concordance overall between the father's face and chosen stimuli, but if the women rated their fathers highly the proportions of the attractive stimuli was more similar to the fathers' faces. They did not, however, test for similarity between the daughters' and the fathers' faces. There is a possibility that the females that rated their fathers most positively also were more facially like their fathers, so a self-referential preference may be a contributing factor. Other studies have found evidence that daughters of older men tend to choose older husbands (*Wilson & Barrett, 1987*), and prefer age cues in male faces compared to cues of youth (*Perrett et al., 2002*). This effect may be the result of a parental tendency, but it could also simply be due to familiarity with older people. In addition, the above studies did not investigate whether the chosen husbands and stimuli had any facial resemblance with the females' fathers or the self.

Although the presence of mirrors, photos, films, and drawings in modern times may have considerably boosted, compared to the evolutionary past, sexual self-imprinting in humans, the empirical evidence in humans for sexual imprinting (either positive or negative) remains weak (e.g., *Marcinkowska & Rantala, 2012*; *Rantala & Marcinkowska, 2011*). Moreover, it remains unclear if self-imprinting should be considered an adaptive strategy or it could have become maladaptive by promoting an excessive tendency towards inbreeding. However, there is empirical evidence suggesting that the 'like mate with like' strategy (whether this may based on kin's or self's phenotypes) can confer adaptive value. Remarkably, it has been shown in a study on the whole population of Iceland that a moderate degree of genetic similarity increases reproductive success and genetic compatibility in humans; specifically, there was a positive association between kinship and fertility, so that Icelandic couples that were mildly related (i.e., third or fourth cousins) had the greatest reproductive success and the highest number of children who further reproduced (*Helgason et al., 2008*). Moreover, self-resemblance in both physical and psychological phenotypes may indicate suitability to one's environment and, in humans, could promote the partners' successful cooperation, relevant for the survival of the offspring (*Godoy et al., 2008*). There are also examples of use in humans of physical cues based on 'kin' resemblance that are strategical in evolutionary terms, for example for detecting cuckoldry (for example in the face, see *Bovet et al., 2012*; *Laeng, Mathisen & Johnsen, 2007*; *Platek & Thomson, 2007*).

The well-documented phenotypic similarity among partners in many human societies may thus reflect inclusive fitness mechanisms, where there can be increased gene duplication without an increase in reproductive investment and a reduced cost

of altruism (*Epstein & Guttman, 1984*; *Thiessen & Gregg, 1980*). Given that phenotypic facial characteristics are known to be highly heritable in humans (*Baydas et al., 2007*; *Weinberg et al., 2013*), faces may be one of the best visual clues to genetic similarity (*Kazem & Widdig, 2013*; *Holmes, 2004*; *Bovet et al., 2012*). Indeed, for both sexes, ratings of facial attractiveness have been found to be a better predictor of general physical attractiveness than ratings on body images (*Currie & Little, 2009*; *Peters, Rhodes & Simmons, 2007*). Whether people could see themselves having a relationship with a stranger can be predicted by the stranger's face attractiveness (*Currie & Little, 2009*), which implies that face attractiveness is an important feature in human mate choice decisions and that faces may provide relevant signals of heritable quality (*Tregenza & Wedell, 2000*).

To conclude, although a universal preference for the human prototype is well-documented (e.g., *Rhodes, 2006*), such a preference could derive from general processing mechanisms or from self's exposure to other non-related individuals. Thus, a key difference from parental or self-referential "imprinting" and general prototype learning is that the latter may be based on self's exposure to non-related and mostly unknown individuals over a lifetime; whereas the former would be linked to early exposure to figures of attachment and self. Thus, mate choice could be the result of a delicate balance between opposing tendencies: inbreeding versus outbreeding; universal "ideals" of beauty versus highly idiosyncratic preferences based on one's own appearance. The interplay between choices based on universal standards of beauty (i.e., symmetry and averageness versus the personal self-based ones) is well expressed in an aphorism by poet *Auden (1962)*: "Narcissus does not fall in love with his reflection because it is beautiful, but because it is his".

### Funding
The study was funded by the University of Oslo through a student grant. The funders had no role in study design, data collection and analysis, decision to publish, or preparation of the manuscript.

### Grant Disclosures
The following grant information was disclosed by the authors:
University of Oslo.

### Competing Interests
The authors declare there are no competing interests.

### Author Contributions
- Unni Sulutvedt conceived and designed the experiments, performed the experiments, analyzed the data, contributed reagents/materials/analysis tools, wrote the paper, prepared figures and/or tables, reviewed drafts of the paper.

- Bruno Laeng conceived and designed the experiments, analyzed the data, contributed reagents/materials/analysis tools, wrote the paper, prepared figures and/or tables, reviewed drafts of the paper, initiated the study and supervised Unni during her master studies.

## Human Ethics

The following information was supplied relating to ethical approvals (i.e., approving body and any reference numbers):

Psychological research in Norway is subject to ethical review by the regional medical research board only if the research involves patients, children or animals and involves drugs, genetic samples or invasive techniques. Since none of these conditions applied to the present study, the academic institution demanded only that the project comply with Declaration of Helsinki guidelines and that informed consent be obtained from the participants. We obtained written consent from all participants. All information was handled and stored anonyomously, and participants were free to withdraw from the project. In addition, participants gave their written informed consent to publication of their photographs.

## Supplemental Information

Supplemental information for this article can be found online at http://dx.doi.org/10.7717/peerj.595#supplemental-information.

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
