# Peer review of "The self prefers itself? Self-referential versus parental standards in face attractiveness"

_PeerJ, doi:10.7717/peerj.595_

## Round 0.1 · original submission · Major Revisions

This study received favorable comments from both reviewers. There were, however, many suggested areas for revision and improvement. I agree with the comments of the reviewers and would like to see them addressed in revision. I also found the manuscript interesting and worthy of publication (though with a major caveat below about the interpretation of the results).

I worry about the confounding effects of age on this study, even though some age corrections were attempted. This is discussed in the context of a preference for mothers’ faces over fathers’, but not as a potential confounding factor in the main conclusion supporting self over parents. I would like to see this caveat removed from the present placement, where the implication is that it refers to attachment to mothers, and discussed as a general caveat throughout the paper (including the abstract). I realize that this makes the results equivocal with regard to a comparison of self vs parent, but unless the authors can convince me otherwise I feel it is a more honest presentation.

Preferences for morphs with younger faces would be consistent with the higher ranking of the both the self and prototype faces (both younger) over the parental morphs (both older) (I’m not sure if these are significant or trends). Importantly the same pattern holds in table 2, where I see no way to account for it other than the influence of age.

I also wonder about the use of makeup in the participants (the participant in figure 1 appears to have makeup on). If the younger participants are more likely to be wearing makeup in their photos than are their older parents this could influence whether they are perceived as more attractive.

Why are the labels in table 1 and table 2 different? E.g., in Table 1 why is “Self” used in the labels for the parents for the female participants but not the male participants? For female participants in Table 2 shouldn’t the second line down read “Male” prototype morph?

·

Basic reporting

The authors should discuss at greater length the evolutionary rationale for self-preference. In particular, they should discuss the relevance of supernormal stimuli with respect to the face template. I also question the use of the term "imprinting" if there is no critical period, as the authors suggest on lines 15-16 "exposure to a range of facial variations during a lifetime."

I also question the authors' belief that we are attracted to "average" faces. There is an abundant literature on the subject of attraction to symmetrical faces, but attraction to average faces is much less documented. There is certainly much evidence to the contrary (see general comments below).

Experimental design

No comments

Validity of the findings

No comments. The findings seem to be correct.

Additional comments

The greatest weakness of this paper is its grounding in evolutionary theory. What is the adaptive value of preferring a mate who resembles oneself? The authors only mention that "some degree of genetic compatibility may be important for reproductive success" (p. 3). In other words, one should avoid potential mates whose genotypes are less suited to one's conditions of life. Because such potential mates would look less similar to oneself than those who share one's conditions of life, it would be adaptive for humans to have a mental algorithm that promotes sexual attraction to similar-looking people.

But was this a significant adaptive problem during human evolution? Until recent times, people generally interacted with other closely related people. The main problem was the risk of mating with someone who was too similar, i.e., inbreeding. Pierre van den Berghe (1983) makes this point when he points to the unlikelihood of a genetic basis for racial discrimination:

"The typical situation is that ethnies which have been neighbors for a while also look alike on the average. More precisely, genetic variation within each group is typically much larger in virtually every phenotype than mean differences between groups. Therefore, racism, in the majority of cases, is no good at all in discriminating between neighbors. Racism would do a much better job of discriminating between distant groups, but, unless they meet, distant groups are seldom interested in discriminating between themselves; indeed, they are often not even aware of each other's existence."

This older social environment would have certainly offered many opportunities for imprinting on faces of similar-looking people, particularly those of family members. Until recent times, however, there would have been much fewer opportunities for self-imprinting on one's own face. Mirrors began to be mass-produced less than two centuries ago, being previously very expensive and limited to the elite. The average person would have more often seen the faces of other family members than his/her own face, particularly during the period of infancy when this facial imprinting presumably occurred.

I suspect that some sort of facial imprinting does occur. It is only in recent times, however, that people have imprinted mostly on their own faces. This tendency is stronger in girls than in boys probably because girls tend to look at themselves in the mirror more often. Previously, boys and girls alike would have imprinted on the facial images of other family members. I also suspect that there may be no critical period, i.e., people prefer whatever facial image they have most often seen during their entire lifetime. If so, use of the term "imprinting" may be inappropriate.

Finally, this imprinting effect is probably noticeable only within a limited range of phenotypes, such as we might see among the Norwegian participants of this study. There is much cross-cultural evidence that people can feel sexually attracted to a facial or body image that lies well beyond the normal range of phenotypes within the local population. A good example is hair extension. Throughout much of sub-Saharan Africa, women lengthen their hair by various artificial means, and such practices predate the first contacts with Europeans (Khumalo, 2008). There thus seems to be a mental algorithm that responds positively to the image of a woman with head hair, one corollary being that longer hair stimulates the algorithm more effectively. In sub-Saharan Africa, this sexual preference has not translated into sexual selection for long-haired women because the high incidence of polygyny reduces sexual selection of women to a minimum (Frost, 2008). Yet this sexual preference prevails even where hair is not naturally long, such as among the Mende of West Africa:

""Big hair," "plenty of hair," "much hair"—West African communities, including Mende, admire a fine head of long, thick hair on a woman. Both these elements are crucial: thickness and length. Thickness equals increase in the number of individual strands, and the length is proof of strength. Growing such luxuriant hair requires a Mende woman's patience and care. Because a man's hair is kept shaved or cut close to the scalp, people say that "men don't have hair." Beautiful hair thus is a distinctly female trait; the more of it, the more feminine the woman" (Boone, 1986, p. 184)

Preference for long female hair is probably due to a mental algorithm that originally focused on infant-specific traits, i.e., soft, pliable, hairless, and fair skin, a pedomorphic face, a higher pitch of voice, and longer, silkier hair. Ajose (2012) notes that most African babies are not born with tightly curled hair; initially, most have loose silky hair. This mental algorithm could more easily influence sexual selection of women in populations where women had to compete for sexual access to men, rather than the reverse (as in tropical "female farming" societies where low paternal investment facilitated polygyny and allowed mated men to return to the marriage market).

I can cite other examples of sexual attraction to "supernormal" stimuli, i.e., phenotypes that do not occur naturally. In different culture areas, women have developed different cosmetics for the purpose of increasing the luminous contrast between facial skin and the color of the lip and eye area (Frost, 2011; Russell, 2009; Russell, 2010). There is also the phenomenon of women changing their hair color to a more eye-catching tint that is either rare, like blonde or red, or totally unnatural, like magenta or blue. Again, there is no reason to suppose that an average appearance most effectively stimulates the mental algorithms that influence sexual attraction. Many if not most of these algorithms appear to have no upper limit (Manning, 1972, pp. 47-49).

I wonder how the authors would explain sexual preference for people who look different, sometimes very different, from one's own ethnic group. How, for instance, would they explain the "white slave trade" of the 8th to 18th centuries, when large numbers of European women, possibly millions, were captured and sold to male clients in Muslim Spain, North Africa, the Middle East, and South Asia (Skirda, 2010). This trade largely took place during a time when Europeans were geopolitically weak and thus unable to impose their standards of beauty, either indirectly, through their socio-economic dominance, or directly, through their mass-production of idealized images of European women, e.g., fashion magazines, pornography, popular films, TV shows, and videos, etc.


References

Ajose, F.O.A. (2012). Diseases that turn African hair silky, International Journal of Dermatology, 51 (supp. S1), 12-16

Boone, S.A. (1986). Radiance from the Waters: Ideals of Feminine Beauty in Mende Art, New Haven and London

Frost, P. (2011). Hue and luminosity of human skin: a visual cue for gender recognition and other mental tasks, Human Ethology Bulletin, 26(2), 25-34.

Frost (2008). Sexual selection and human geographic variation, Special Issue: Proceedings of the 2nd Annual Meeting of the NorthEastern Evolutionary Psychology Society. Journal of Social, Evolutionary, and Cultural Psychology, 2(4),169-191. http://137.140.1.71/jsec/articles/volume2/issue4/NEEPSfrost.pdf

Khumalo, N.P. (2008). On the history of African hair care: more treasures await discovery, Letter to the Editor, Journal of Cosmetic Dermatology, 7, 231.

Manning, A. (1972). An Introduction to Animal Behaviour, 2nd edition, London: Edward Arnold.

Russell, R. (2010). Why cosmetics work. In Adams, R., Ambady, N., Nakayama, K., & Shimojo, S. (eds.) The Science of Social Vision. New York: Oxford.

Russell, R. (2009). A sex difference in facial pigmentation and its exaggeration by cosmetics. Perception, 38, 1211-1219.

Skirda, A. (2010). La traite des Slaves. L'esclavage des Blancs du VIIIe au XVIIIe siècle, Paris, Les Éditions de Paris Max Chaleil.

van den Berghe, P. (1983). Class, Race and Ethnicity in Africa, Ethnic and Racial Studies, 6, 221-236.

·

Basic reporting

I like the design of the research, as it completes the existing pool of studies on assortative mating, sexual imprinting and self-referential matching.
Information was presented in a good way in Introduction and Discussion - no pointless overlapping nor repetition.
Authors list studies supporting sexual imprinting in both sexes, but do not write about studies that showed lack of it (Marcinkowska&Rantala 2012 for similarity between parent and partner and Marcinkowska, Moore & Rantala 2013 for prefence towards sibling smilar facs – I propose these two as I know them best, but there is more).
My biggest doubt considers statistical analysis. It seems that authors run alno some of possible (and not all usefull analysis/comaprisons – listed below).
Also there is not enough about aversion in discussion (it is mentioned in the introduction).
I think the data gathered in this study could bring more interesting outcomes if better depicted/analysed.

Experimental design

The findings presented in the article are novel. Research question is clear and well described. It was conducted according to national ethical standards of the country where it was realised.
How many slides the participants saw? It is unclear how the stimuli were presented.
Description of morphing is good.

Validity of the findings

Conclusions are connected with the investigated questions.

Additional comments

suggest using other sex rather than opposite sex (but it is only my personal point of view).
Author use both past and present time in the introduction – it should be unified.
There are couple of repetitions in the text (some listed below), what increases the amount of text, but does not bring any useful background.
There is are no comments to Other condition in Discussion. Would it be possible to compare family and other condition? It would be really useful as as then we would have a blend of prototype + 1face morph in all cases (not comparing prototype and prototype +1 face). Also comparing self-mother with other mother would naturally exclude age confound in the comparison of attractiveness.

Particular comments:
Abstract
L3: may be change to IS - there is no sense in aiming to see whether something may be or may not be.
L6: I suggest rephrasing the sentence to make it easier to read:" ...image manipulations of facial photographs of participants, their mother and father."
L9: "Attractiveness ranks...blends" as the reader is not yet familiar with the analysis (ranks, not e.g. Likert scale) I would suggest not using "rank" here. Maybe attractiveness perception or judgement?
L10: " A clear...found." From here it is not clear whether standard means a process or a stimuli. If it is stimuli then it would be less misleading to call it self-standard and parental standard). If it is a mechanism, than it cannot be preferred. Mechanism can be depicted, shown or used.

Introduction

L9: adaptation F mate choice?
L10-L13: across species – maybe better to change to different primates? „...but reflect innate... change to but IT reflectS innate... . Maybe it is worth to add to this sentence that you speak about the process that affects attractiveness perception in adult life? Also it might help the readers to add here that there are two unrelated processes: 1) sexual imprinting that has a sensitive phase (information lacing) and 2) mere exposure effect, with no sensitive phase.
L18-20: it would be worth adding here that presented studies are all based on non-human animals?
L22: provide reference for seeking family resemblance.
L27-29: To make the paragraph easier to read this sentence should go at the end: … heritable quality. What more, phenotypic facial...
L40: place reference at the end of the sentence.
L43-L45: equal attractiveness can constitute to the similarity, but it does not increase similarity directly. I suggest rephrasing the sentence to make it clear.
L58: of self's partners – in my opinion individual's parents or one's parents sound better
L59: delete “self”
L63: simply IS reduceD
L69: Nojo, Tamura and Ihara is a very important reference to this paper and it is not listed in the reference list
L75: if THEY had been overprotected...?
L89: In present study... I suggest this paragraph should go a the end o the introduction
L104: IN the present study... , believe – I would change it to we POSTULATE
L107: non-sexual preference – does it mean sympathy? Or judging attractiveness of same-sex picture when heterosexual? Maybe it would b useful to explain this better.
L112: the sentence does not bring anything new, and is very similar to line 100 and line 89.
L118-L138: this I very detailed description of methods, s maybe it would be better if it was in materials and methods section of the article.
L123: ...like signal detection analysis – provide explanation
L128: ...mount of likeness... - I am not sure whether authors use likeness and attractiveness for the same preference. If yes, maybe it would be easier to follow if only one word was used. If not, provide definitions on both in text (differences between them?)
L139: if mere exposure effect would be present, wouldn't we expect mother and father preference>self?
L149-L151: this not supported by Marcinkowska & Rantala 2012.
L151: AS this could have...
L152: delete “so”

Materials and Methods

L155-158: add that participants were heterosexual
L171-172: didn't this action influence the attractiveness of pictures, if only some were corrected, and others no? If such procedure was introduced, shouldn't it be eqal treatment for all stimuli?
L173: from THE TIME when children were young
L177 same camera, in standard pose and WITH neutral expression.
L196: what does pseudo-randomized mean?
L204: out of curiosity – did authors measure response time? It could be a good measure
of subconscious response.
L207: why weren't the participants who understood the design of stimuli not excluded from the analysis?
L212-228: why does this supposedly exclude EMBU? Authors do not write analyse only differences between men and women, but rejected women to not rejected women etc. The explanation on EMBU exclusion is unclear for me.
L232: ...there was no significant...”it is unclear to me where was this difference depicted? Was such comparison in the ranking? It is not clear about how the stimuli were presented.

Discussion

L258: imprinting of self's face – I would rather say self-referent or preference for self-similarity.
L269: likeness – did authors mean similarity?
L276: highly – highly what?
L283-286: from the description of the methods I do not see where did authors measure this directly?
L290: and stimuli – what stimuli (whose face)? Unclear
Table1: authors use both self mother and mother – are these two different types of stimuli for different sexes? If not, it should be unified.
Table2: shouldn't it be male prototype morph for female observers?

---

## Round 0.2 · Major Revisions

I have read over the manuscript by Sulutvedt and Laeng. While the responses to the reviewers seems sufficient there is still significant confusion in the Results section (some of this, I believe, was introduced upon revision). This all still has to be clarified before the paper is ready for publication.

I am especially concerned that there is still vagueness over whether there is an age effect. See the comment below for pg 16 line 18. Right now I am reading your data as support for an age effect, at least in female participants (I am perfectly willing to be convinced that I am misinterpreting this, please be clear about this point in your reply). Please consider very carefully whether you can say that there is a preference for self OVER parent given that your data, at least for female participants, seems to support that there is an age effect. If you don't think that you can say there is a preference for self OVER parent, please make it clear that you actually can't distinguish this over age when you say that this is the first study to make that comparison.

I also have several other suggestions, the most significant of which are marked below with an asterisk.

Pg 3, line 10 - I would phrase this as "but the potential for it clearly predates"

*Pg 3, line 12 - "A key assumption" - I don't see how this is really an assumption. Self-imprinting (especially modern self-imprinting) could occur even if it were maladaptive. This study supports that it occurs, but really makes no assumption whether it is adaptive or not. I would rephrase this to remove the suggestion that this in an assumption. In fact, discussion of the adaptive nature of self-imprinting could potentially be moved to the Discussion (although it is fine to leave it here if you prefer it as well).

Pg 6, line 4 - "self's own face" - would this be "during childhood" still? It was only tested with self adult faces, so this sentence should be revised to move self out of the childhood clause.

Pg 6, line 20 - "separately these hypotheses" - which separate hypotheses? That self is preferred at all and that parents are preferred at all? You see "examined phenotypic likeness to oneself versus parents", which specifies a direct comparison (though your subsequent description of this study doesn't). Please clear this up.

Pg 7, line 8 - "resembled kin" - do you mean self? This sounds like you mean self.

Pg 7, line 22 - "self-morph" - with what? In this sentence do you mean self mixed with opposite sex prototype is preferred to actual partner mixed with opposite sex prototype? Please be more specific.

Pg 8, lines 14-18 - this is repetitive with the paragraph above

Pg 10, line 2 - do you mean "no positive change"?

Pg 11, lines 8-11 - this paragraph is also repetitive.

*Pg 15, lines 6-10 - This seems odd given the ranks, was the variance in the father's morph particularly high? Also in line 9 I don't see how no different provides support for attraction. That is, I think you're implying that that the absence of a negative response provides support for attraction, but they're really not the same thing.

*Pg 15, line 19 - "another young female" - should this be male?

*Pg 15, line 20 - "also significant" - why "also"? The previous comparison discussed was not significant.

*Pg 16, line 1 - "females' preference" - do you mean that it prevented the female participants from liking the morph less? This could be clarified, but again I don't think you can exactly say they are preferred. Also if this is what you mean then you can't really sat that it controls for an age effect.

*Pg 16, line 1 - "Finally, self" - should this read "other"?

* Pg 16, line 5 - "a difference in ranks" - but this is not significant, correct?

* Pg 16, line 18 - but isn't the other morph a female morphed with the prototype while the other's father morph is a male morphed with the prototype? The different could be based on sex, not age. Was the other versus other's mother comparison significant? This one would suggest age. **Also the comparison of other versus other's father in the female participant section would provide strong evidence for an age effect, since there is no other way to explain a difference in that comparison. Was this significant? If so, please point out that you have evidence for an age effect and temper all remaining mentions of support for self-imprinting versus parental imprinting accordingly.

Pg 17, line 11 - "for the male participants the prototype" - this only makes sense in comparison with the fact that the prototype is superior to other faces - make that connection here (also see the comments on this point above).

---

## Round 0.3 · accepted · Accept

The authors did an excellent job with this round of revision. The ms is much more straightforward and clear. I appreciate the author's willingness to respond to comments so thoroughly.

I did catch just one minor error - the phrase "such a morph" is missing on Pg 13, line 17 it was specified to be there in the cover letter but is missing in the actual file). Please contact the editorial office about fixing this.